# Association between Clinical and Histopathological Findings in Intestinal Neuronal Dysplasia Type B: An Advance towards Its Definition as a Disease

**DOI:** 10.3390/life13051175

**Published:** 2023-05-12

**Authors:** Anderson Cesar Gonçalves, Isabelle Stefan de Faria Oliveira, Pedro Tadao Hamamoto Filho, Erika Veruska Paiva Ortolan, Simone Antunes Terra, Maria Aparecida Marchesan Rodrigues, Pedro Luiz Toledo de Arruda Lourenção

**Affiliations:** Botucatu Medical School, São Paulo State University (UNESP), Botucatu 18618-687, Brazil

**Keywords:** intestinal neuronal dysplasia type B, constipation, children

## Abstract

Introduction: Intestinal neuronal dysplasia type B (IND-B) is a controversial entity that affects the submucosal nerve plexus of the distal intestine. The lack of definition of the causal relationship between histological findings and clinical symptoms has been identified as the primary point to be elucidated in the scientific investigation related to IND-B, which is essential for it to be considered a disease. Objective: To investigate the relationship between histopathological findings and symptoms in a series of patients with IND-B. Methods: Twenty-seven patients with histopathological diagnosis of IND-B, according to the Frankfurt Consensus (1990), who underwent surgical treatment through colorectal resections were included. Data from medical records regarding the clinical picture of the patients at the time of diagnosis, including the intestinal symptom index (ISI) and a detailed histopathological analysis of the rectal specimens, were retrieved. Exploratory factor analysis was performed, applying the principal components method for clusters with Varimax rotation. Results: Two factors were determined: the first, determined by histopathological and clinical variables, and the second, composed of the main symptoms presented in patients with IND-B, including ISI. Factorial rotation showed the association between the two factors and, through a graph, demonstrated the proximity between ISI values and histopathological alterations. Conclusion: There was evidence of an association between the clinical features presented by patients with IND-B and the histopathological findings of the rectal samples. These results support the understanding of IND-B as a disease.

## 1. Introduction

Intestinal neuronal dysplasia (IND) is a condition that affects the intestinal nerve plexuses. It is classified as part of a heterogeneous group of complex diseases that affect the enteric nervous system [1,2]. Two forms of IND are described: IND type A (IND-A) and IND type B (IND-B) [3]. Less than 5% of all IND cases involve IND-A, which is unusual. Individuals with IND-A frequently exhibit symptoms during the newborn period, ranging from acute intestinal obstruction to diarrhea with hemorrhagic stools. IND-A is characterized by hypoplasia or aplasia of the adrenergic enteric nervous system [4,5]. In contrast, more than 95% of cases are represented by IND-B, which is why this type has received more attention in the literature and why many authors use the term IND-B as synonymous with IND [2]. The hyperplasia of the parasympathetic submucosal plexuses is a hallmark of IND-B. Hyperganglionosis, large ganglia, ectopic ganglion cells, and enhanced acetylcholinesterase activity in the lamina propria and around the submucosal blood vessels are typical histological IND-B characteristics. The IND-B alterations are more prevalent in the distal colon; however, they may be present in the enteric nervous systems of other intestinal segments. They may also happen in individuals of every age, from newborns to adults, alone or in conjunction with Hirschsprung’s disease (HD) [5,6,7]. IND-B can cause severe constipation that is unresponsive to clinical management, soiling and hemorrhagic stools, acute bowel obstruction, or enterocolitis episodes. IND-B symptoms occasionally mimic those of HD, which is its primary differential diagnosis. It has been proposed that IND-B segments upstream of a part of aganglionosis may contribute to the persistence of gastrointestinal issues after HD surgical treatment. Between 6% and 44% of patients with HD may experience this association [6,7,8,9].

IND-B incidence was estimated to be 1 per 7500 births [10]. However, the reported rates range from 0.3% to 40% of all rectal suction biopsies [6,11,12,13]. The lack of consensus on the histopathological diagnostic criteria may be the reason for this substantial heterogeneity [4,7].

The diagnosis of IND-B fundamentally depends on the histopathological analysis of rectal biopsies [1,2,14]. However, the morphological criteria for its diagnosis have changed over the years, exposing the lack of consensus on the disease and making it difficult to compare studies carried out [6,12,15,16]. Hyperplasia of the submucosal nerve plexus is the morphological finding that defines IND-B; however, it is differently characterized by the criteria [6,12,15,16].

The pathogenesis of IND-B is still not fully understood more than 50 years after its initial description [17]. Commonly, IND-B is associated with clinical symptoms of chronic intestinal constipation that affect children in their first years of life, such as those of HD; however, IND-B has histopathologic features characterized by hyperplasia of the submucosal nerve plexus [14]. Despite the intense scientific research performed in the last decades, which includes more than 250 published scientific articles, there are still gaps in our knowledge of IND-B’s definition, pathogenesis, diagnostic criteria, and therapeutic possibilities [2,4,6,7].

Two therapeutic modalities have been used in patients with IND-B: conservative clinical therapy and surgical treatment [7,18,19]. Conservative therapy is composed of changes in diet, laxatives, and enemas in cases of fecal retention [4,19,20]. Surgical treatment includes sphincterotomy, extensive surgical resection, or temporary colostomy [21,22,23,24,25]. The current recommendation is that patients with no complications receive conservative treatment [4,7,14,19].

One of the main controversial points in the study of IND-B concerns the existence or not of a cause–effect relationship between its histopathological findings and clinical symptoms [7,15,16]. In most cases, the IND-B diagnosis is based on the histopathological analysis of rectal biopsies from patients with constipation, which is usually refractory to clinical treatment. A minority of cases may present with acute intestinal obstruction or enterocolitis [7]. On the other hand, histopathological changes compatible with the diagnosis of IND-B were found in the colon segments of 36 completely asymptomatic children [26]. Other studies have failed to demonstrate a direct correlation between histopathological findings, clinical symptoms, and radiological and manometric changes [23,24,25,26,27,28,29]. Faced with these controversies, several authors do not consider IND-B as a true clinical entity but only as a histopathological alteration of the enteric nervous system that may or may not present symptoms [16,30,31].

The lack of definition of the causal relationship between histological findings and clinical symptoms has been identified as the primary point to be elucidated in the scientific investigation related to IND-B. This is the main link still pending for the accurate characterization of IND-B as a disease and the definitive establishment of its diagnostic criteria and therapeutic possibilities [2,7,14,15,16,30].

Thus, this study aimed to investigate the association between the histopathological findings identified in patients with IND-B and the clinical symptoms present at the time of diagnosis.

## 2. Materials and Methods

A single-center, retrospective, uncontrolled, descriptive, and analytical study was performed. Patients aged up to 15 years old with the diagnosis of IND-B established through histopathological analysis of rectal biopsies, according to criteria proposed in the Frankfurt Consensus (1990), and who had undergone surgical treatment through extensive colorectal resections between 1998 and 2012 were included [32]. Patients who refused to participate in the study and those with any other type of associated gastrointestinal neuromuscular disease were excluded. The local Research Ethics Committee approved this study (protocol number CAAE 80018217.1.0000.5411).

We retrieved information from the medical records of patients eligible for the study. We obtained the following clinical data of interest: age at onset of symptoms, age at diagnosis, sex, gestational age, birth weight, the clinical course in the neonatal period, delay in meconium passage, presence of associated malformations, history of acute intestinal obstruction, and the reason for surgical indication. In addition, we obtained specific information on bowel habits present at the time of diagnosis of IND-B, mainly related to symptoms of constipation, including bowel movement frequency, stool consistency, painful or straining bowel movements, bowel bleeding, abdominal pain, the presence of fecalomas, need for bowel washing, and fecal incontinence. These data allowed us to assess the presence of each of the six Rome IV criteria (RIV C) [33,34] for the diagnosis of constipation in children (RIV C1: ≤2 bowel movements per week; RIV C2: >1 episode of fecal incontinence per week if toiled trained; RIV C3: retentive posturing or excessive volitional stool retention; RIV C4: painful or hard bowel movements; RIV C5: large fecal mass present in the rectum; RIV C6: history of large diameter stools that can obstruct the toilet) and also to apply the intestinal symptom index (ISI), which is an index proposed to summarize the most common signs and symptoms in the clinical course of IND-B [18,19].

We also retrieved data from a previous study that performed a detailed histopathological analysis of surgical specimens from these patients [35]. This analysis was performed from histological sections processed by standard histology (hematoxylin and eosin) and calretinin immunohistochemistry, analyzed jointly by two experienced pathologists without knowledge of the clinical information of the patients. We access data from the descriptive analysis of the submucosal nerve plexus, including the presence of ganglia with signs of immaturity, the size and morphology of the neurons in the nerve plexus, and the expression of calretinin in neurons in the plexus, in mucosal nerve fibrils, and in heterotopic neurons in the lamina propria or in the muscularis mucosa. In addition, we access data from the quantitative morphometric analysis of the submucosal nerve plexus, including the number of neurons in 25 ganglia, the maximum number of neurons per ganglion, the number of fields required to identify 25 ganglia, the maximum thickness of a nerve trunk (μm), the area of a neuron (μm²) and the number of giant ganglia (with more than eight ganglion cells) in 25 analyzed ganglia [35].

The data were submitted for descriptive analysis, allowing for the clinical and pathological characterization of the patients included in the study. Descriptive analysis was performed using absolute numbers and respective percentages and determining the mean ± standard deviation, and median (minimum/maximum) values.

In order to investigate the relationship between the clinical findings at the time of IND-B diagnosis and the results of the histopathological analysis of the rectal samples, we performed an exploratory factor analysis, applying the principal components analysis method for clusters with varimax rotation. This multivariate analysis seeks to identify which of the measured variables is most important to determining a phenomenon. This analysis converts a set of possibly correlated data into a set of linearly uncorrelated variables from which the main components originate. A good data correlation between two factors means that both are intrinsically linked [36,37]. In the present analysis, we did not perform the Kaiser–Meyer–Oklin (KMO) and sphericity tests because the number of variables was greater than the number of observations. We established a cut-off point of 0.5 for factor loadings. We performed the distribution of the scores of the determining factors using a score chart, enabling the detection of clusters, outliers, and trends [36,37].

The significance level was 5%, and the analysis was performed in the SAS software for Windows v. 9.4.

## 3. Results

Twenty-nine patients, aged between 0 and 15 years old, had received the diagnosis of IND-B, established through histopathological analysis of rectal biopsies, and subsequently underwent surgical treatment through extensive colorectal resections. Two patients were excluded for not agreeing to participate in the study. Thus, 27 patients were included in the study.

### 3.1. Clinical and Demographic Data

Table 1 presents the primary demographic and personal background information, focusing mainly on the neonatal period. The mean age of patients at the time of surgery was 26.71 ± 38.7 months, with a median of 4 months (0.06 to 172 months). The main clinical picture presented was constipation in 20 patients (74.1%), intestinal obstruction (Figure 1A) in 6 patients (22.2%), and enterocolitis in 1 patient (3.7%).

Table 2 presents the main clinical variables related to the patient’s bowel habits and abdominal symptoms at the time of the IND-B diagnosis. Patients had, on average, 2.26 ± 3.16 bowel movements per week, with a median of 1 (0–14) bowel movements per week. The mean maximum number of days without bowel movements was 9.07 ± 6.24, with a median of 7 (1–25) days. Except for patients who had an intestinal obstruction in the neonatal period and one patient who had incomplete information in the medical record, all the other 23 patients met the Rome IV criteria for the diagnosis of constipation [17,18]. The pre-treatment ISI had a mean of 3.47 ± 1.34 and a median of 3.0 (1.0/6.0) [19,20]. All patients were treated surgically: 21 by the transanal endorectal pull-through technique, three by colostomies and the subsequent Soave pull-through technique, and three by colostomies following the Duhamel pull-through technique [38,39,40]. Fifteen patients (55.6%) underwent surgical treatment in the first year of life; 9 (33.3%) were between 1 and 5 years old, and three (11.1%) were over six years old.

### 3.2. Histopathological Analysis of Rectal Specimens

Table 3 presents the statistical analysis of the data obtained in the quantitative morphometric analysis of the surgical specimens of the rectum processed by hematoxylin and eosin. Table 4 shows rectal specimens’ descriptive morphological analysis using hematoxylin, eosin, and calretinin immunohistochemistry. The submucosal nerve plexus showed alterations in all cases, with hyperplasia of the submucosal nerve plexus characterized by hyperganglionosis and hypertrophy of the parasympathetic nerve trunks (Figure 1B) [35]. There was immunostaining for calretinin in the neurons of the submucosal nerve plexus in all cases analyzed. Most patients had ganglia with signs of immaturity, characterized by vacuolization with small and anisomorphic neurons [35].

### 3.3. Factor Analysis

We performed an exploratory factor analysis applying the principal components method, which allowed us to correlate the preoperative clinical data and the histopathological examination results. Two factors were determined, with 28% of variation explained by Factor 1 and 17% for Factor 2, totaling 45% of variation explained by these factors (Appendix A). Applying a cut-off point of 0.5 for the factor loadings, we determined the main variables that were composed of two factors: -Factor 1, composed of the variables: the number of giant ganglia; the number of neurons in 25 ganglia; the mean number of neurons per ganglia; the maximum number of neurons per ganglia; ganglia with signs of immaturity; problems in the neonatal period; large diameter stools that can obstruct the toilet; less than two bowel movements per week; rectal bleeding; and the need for bowel washouts.-Factor 2, composed of the following variables: intestinal symptom index (ISI); history of fecal retention; painful or hard bowel movements; abdominal distention; abdominal pain; at least one episode of fecal incontinence per week; and being born to term.

Then, a varimax factorial rotation was processed, which is a process of adjusting the factor axes to obtain a more straightforward and pragmatically significant factor solution whose factors are more easily interpretable [22,23]. Figure 2 shows the graphic distribution of the result of this process, with the scores of the second factor versus the scores of the first factor. Values are standardized such that the mean is zero, and the distance between scores is measured in standard deviation. This analysis makes it possible to detect clusters, outliers, and trends. The upper right quadrant of Figure 2 indicates variables that lead in the same direction (the means are above zero for both factors: positive direction x positive direction), indicating that the variables present in this quadrant have similar effects [36,37]. Thus, we observed the presence in this quadrant of the score referring to the Intestinal Symptom Index and of the scores referring to the main histopathological alterations of the rectal samples, including the mean of neurons per ganglia, the number of neurons in 25 ganglia, the number of giant ganglia, and the presence of ganglia with signs of immaturity.

## 4. Discussion

Most of the patients with IND-B included in this study are male, born at term, and have not presented other clinical problems in the neonatal period or other associated congenital malformations. Most of these patients started showing symptoms already in the neonatal period, many with a delay in meconium passage. Seven patients required urgent surgeries due to acute intestinal obstruction or enterocolitis. The other 20 patients underwent elective surgery due to severe constipation refractory to conventional clinical treatments. During the analyzed period, due to the lack of consistent guidelines on the treatment of IND-B, the modalities of treatment for patients with severe constipation who were diagnosed with IND-B based on rectal biopsies were chosen at the surgeons’ discretion. Some surgeons opted for elective surgical treatment with colorectal resections. Others opted for conservative treatment with diet, laxatives, and enemas [19]. Regarding the histopathological analysis of the surgical specimens, all patients presented the diagnosis of IND-B according to the Frankfurt Consensus criteria [32,35]. They also showed immunoexpression for calretinin in the neurons of the plexus [35].

Exploratory factor analysis allowed the investigation of the relationship between the patient’s clinical symptoms and the histopathological characteristics found in the rectal samples. This analysis determined two main factors capable of representing almost half of the explained variations. Factor 1, responsible for most of the described variations, was composed of histopathological variables and some clinical variables that can be considered severe ones, such as the need for bowel lavage and rectal bleeding. Factor 2, on the other hand, was composed of the main clinical symptoms presented by patients with IND-B, including the intestinal symptom index (ISI), an index proposed to summarize the most common signs and symptoms in the clinical picture of IND-B [18,19]. 

After the factorial rotation procedure, it was possible to identify an association between the main components of these two factors through a graphic representation (Figure 2). In this graphic, the values are standardized such that the mean is zero and the distance between scores is measured in standard deviation. The factors in the upper right quadrant are positively correlated (the means are above zero for both factors). Thus, close relationships between the Intestinal Symptom Index—ISI (legend “P”) and the histopathological alterations classically associated with IND-B, such as the number of neurons in 25 ganglia (legend “W”) and the mean of neurons per ganglia (legend “Y”), became evident. These components lead in the same direction in the graph, contributing to the same clinical meaning in the investigated relationships between symptoms and histopathological alterations. These findings support the understanding that there is an association between clinical symptoms and histopathological findings in patients with IND-B. This reinforces the theory that IND-B is a disease with specific symptoms and determined organic changes in the gastrointestinal tract.

Some limitations must be considered, particularly those related to the retrospective design, the use of paraffin-embedded samples with different storage periods and clinical information obtained from medical records, the limited sample size, and the absence of a control group. Intestinal neuronal dysplasia Type B is a rare disease. In addition, the uncertainties related to its definition and diagnostic histopathological criteria certainly influence the prevalence estimates, which may be even lower [4,7,10]. Thus, although our sample is numerically limited, it is one of the most significant described in studies that evaluated pathophysiological aspects of IND-B. Moreover, determining a reliable control group composed of colon samples from healthy children without intestinal symptoms is one of the most significant challenges for scientific investigations related to IND B. Obtaining intestinal samples for use as controls has ethical limitations due to the need to perform rectal biopsies, which are invasive procedures [9]. However, the statistical model used in our study mitigated the limitation related to the lack of a control group, making it possible to precisely assess the presence of an association between the histopathological characteristics in the rectal samples and the symptoms presented by the same patients.

Another limitation of the study is related to the IND-B itself and the numerous uncertainties surrounding this pathological entity. All patients in the study had received the histopathological diagnosis of IND-B based on the Frankfurt Consensus criteria [32]. Although these morphological criteria are the most reported in scientific publications on IND-B, many authors have defended the need for quantitative criteria related to the minimum number of neurons present in the submucosal nerve plexus for the diagnosis [6,23,27,35,41,42,43]. Our study could not use these quantitative criteria because they depend on fresh material for freezing sections and specific histochemical reactions [6,43]. However, the morphological characteristics that define the IND-B, regardless of the histopathological criterion used, represented by hyperplasia of the submucosal nervous plexus, were present in all analyzed cases [32,35,44,45]. The uncertainty related to the definition of IND-B as a disease is one of the most controversial points in the most recent publications on intestinal dysganglionosis in children [2,7,14,15,16,30]. In a recent review, Kapur & Reyes-Mugica (2019) concluded that IND-B continues to be considered an undefined histopathological phenotype with uncertain clinical relevance and that it is imprudent to make clinical decisions based on this histopathological diagnosis [16]. However, in medical practice, we continue to find children with severe constipation or intestinal obstructions who undergo rectal biopsies for diagnostic workup for Hirschsprung’s disease, presenting ganglion cells distributed in an abnormal pattern in the nervous plexus of the submucosa with signs of hyperplasia. Schmittenbecher et al. (2000) consider that these histological alterations determine the diagnosis of IND-B, which should be viewed as a clinical entity since the symptoms presented by these patients are evident and often remain for several years, influencing their quality of life [46]. These authors defend the idea that patients with IND-B should be treated in a specific way and highlight the need to develop specific treatment algorithms [46]. Most of the experts present at the 4th International Symposium on Hirschsprung’s Disease and Associated Neurocrystopathies, which took place in Genova, Italy, in 2004, and most participants in a recent survey by the European Association of Pediatric Surgeons claim to believe in the existence of IND-B, justifying the performance and need for studies on this disease [47,48].

## 5. Conclusions

Our study presents evidence of the association between the clinical signs and symptoms and the alterations found in the histopathological analysis (morphological and morphometric) of the rectal samples of patients with IND-B. The disease concept is based on disturbances in the functions of an organ, the psyche, or the organism, associated with specific signs and symptoms [49]. These relationships, proven in this series of cases of patients with IND-B, between symptoms and organic alterations of the gastrointestinal tract seem to be the link for the understanding that IND-B is a disease, justifying further research to investigate aspects of its etiopathogenesis and the need to improve its methods and diagnostic criteria, as well as to establish a specific effective treatment.

## Figures and Tables

**Figure 1 life-13-01175-f001:**
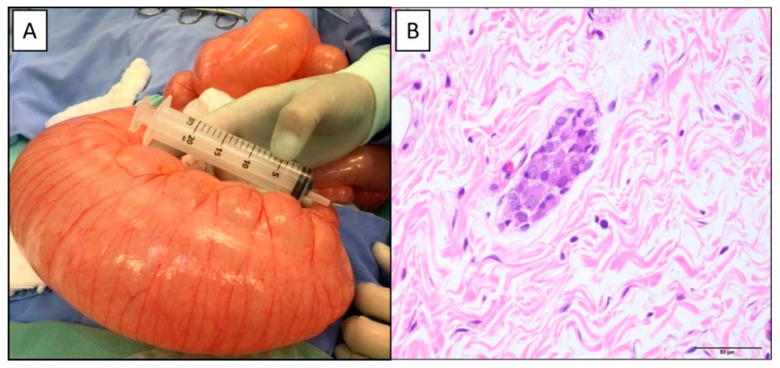
A 2-year-old patient with intestinal neuronal dysplasia type B who developed acute intestinal obstruction: (**A**)—intraoperative with significant distention of the rectum and sigmoid; (**B**)—histopathological analysis with submucosal nerve plexus with hyperganglionosis in the rectum (H&E, 400×).

**Figure 2 life-13-01175-f002:**
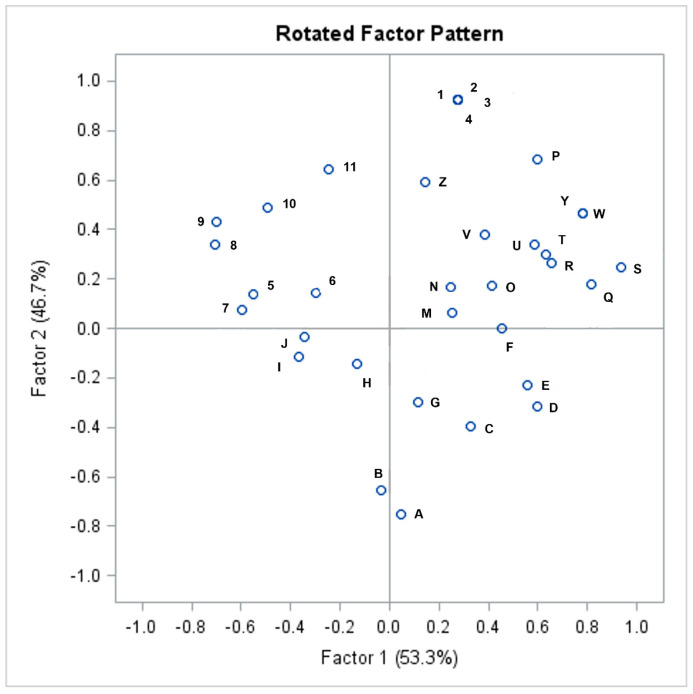
Graphic distribution of the factorial rotation process: A—Maximum number of days without bowel movement; B—Calretinin expression in heterotopic neurons in the muscularis mucosa; C—Birth weight; D—RIV C1: < 2 bowel movements per week; E—Need for bowel washouts; F—Bud-like nerve cell groups along nerve trunks; G—Small neurons; H—Maximum width of a plexus; I—Sex; J—Calretinin expression in nerve fibers in the lamina propria of the mucosa; M—Delayed meconium passage; N—Other malformations; O—RIV C6: large diameter stools that can obstruct the toilet; P—Intestinal Symptom Index (ISI); Q—Number of giant ganglia; R—Maximum number of neurons per ganglia; S—RIV C5: large fecal mass present in the rectum; T—Problems in the neonatal period; U—Rectal bleeding; V—Ganglia with signs of immaturity; W—Number of neurons in 25 ganglia; Y—Mean of neurons per ganglia; Z—Born to term; 1—RIV C3: history of episodes of fecal retention; 2—RIV C4: painful or hard bowel movements; 3—Abdominal distension; 4—Abdominal pain; 5—Number of plexus with more than 7 neurons; 6—Age of onset of symptoms; 7—Anisomorphic neurons; 8—Area of a neuron; 9—Age at diagnosis; 10—Calretinin expression in heterotopic neurons in the lamina propria; 11—RIV C2: >1 episode of fecal incontinence per week.

**Table 1 life-13-01175-t001:** Demographic variables and the personal history of patients in the study.

Demographic Variables and Personal Background		n	%
Sex	Male	18	66.7
	Female	9	33.3
Gestational age	Full-term	20	74.1
	Premature	7	25.9
Birth weight	<3 kg	13	50.0
	≥3 kg	13	50.0
	Present	9	34.6
Problems in the neonatal period	Absent	17	65.4
	no information	1	-
	Present	1	3.80
Other congenital malformations	Absent	25	96.2
	no information	1	-
	Neonatal	20	76.9
Age group at the onset of symptoms	Infant	4	15.4
	>2 years	2	7.70
	no information	1	-
	Present	11	44.0
Delayed meconium passage	Absent	14	56.0
	no information	2	-

n: number of patients; %: percentage distribution.

**Table 2 life-13-01175-t002:** Clinical variables related to bowel habits and abdominal symptoms at the time of IND-B diagnosis.

Clinical Variables		n	%
RIV C1: ≤2 bowel movements per week	Present	21	91.3
	Absent	2	8.7
	no information	4	-
RIV C2: >1 episode of fecal incontinence per week	Present	6	60.0
	Absent	4	40.0
	no information or aged <3 years	13	-
RIV C3: retentive posturing or excessive volitional stool retention	Present	22	95.6
	Absent	1	4.4
	no information	4	-
RIV C4: painful or hard bowel movements	Present	23	95.8
	Absent	1	4.2
	no information	3	-
RIV C5: large fecal mass present in the rectum	Present	7	30.4
	Absent	16	69.6
	no information	4	-
RIV C6: history of large-diameter stools that can obstruct the toilet	Present	5	21.7
	Absent	18	78.3
	no information	4	-
	Present	20	86.9
Abdominal distension	Absent	3	13.1
	No information	4	-
	Present	19	82.6
Abdominal pain	Absent	4	17.4
	no information	4	-
	Present	6	26.1
Rectal bleeding	Absent	17	73.9
	no information	4	-
	Present	19	73.1
Need for bowel washouts	Absent	7	26.9
	no information	1	-

n: number of patients; %: percentage distribution.

**Table 3 life-13-01175-t003:** Quantitative morphometric analysis of rectal specimens processed by hematoxylin and eosin.

Quantitative Morphometric Variable	Mean	Standard Deviation	Median	Minimum	Maximum
Number of plexuses with more than 7 neurons	3.62	2.09	3.0	1.0	8.0
Number of giant ganglia	1.85	1.83	2.0	0	7.0
% of giant ganglia	7.4	7.33	8.0	0	28.0
Number of fields to locate 25 ganglia	17.96	2.42	18.0	13.0	22.0
Number of neurons in 25 ganglia	92.0	21.64	96.0	51.0	130.0
Maximum number of neurons per ganglia	10.7	4.29	10.0	5.0	20.0
Maximum width of a plexus	42.99	9.38	45.13	24.03	61.9
Area of a neuron	101.4	35.61	96.59	57.92	239.75
The mean of neurons per ganglia	3.68	0.86	3.84	2.04	5.2
Number of giant ganglia per mm²	0.50	0.56	0.42	0	2.49

**Table 4 life-13-01175-t004:** Descriptive morphological analysis of rectal specimens processed by hematoxylin-eosin and immunohistochemistry for calretinin.

Morphological Findings (Hematoxylin-Eosin)	n	%	Immunohistochemical Expression for Calretinin		N	%
Bud-like nerve cell groups along nerve trunks	6	22.2	Nerve fibers in the lamina propria of the mucosa	focal expression (+) ^a^	11	40.7
				focal expression (++) ^b^	13	48.1
				diffuse expression (+++) ^c^	3	11.1
Ganglia with signs of immaturity	20	74.1	Heterotopic neurons in the muscularis mucosa		11	40.7
Large neurons	6	22.2				
Normal-sized neurons	18	66.7				
Small neurons	20	74.1	Heterotopic neurons in the lamina propria		14	51.9
Anisomorphic neurons	16	59.3				

n: number of patients; %: percentage distribution; ^a^ focal expression (+) visualized only at ×400 magnification; ^b^ focal expression (++) visualized from ×200 magnification; ^c^ diffuse expression (+++) visualized from ×200 magnification.

## Data Availability

The data presented in this study are available on request from the corresponding author. The data are not publicly available due to privacy and or ethical restrictions.

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
