# Peer review of "Association between Clinical and Histopathological Findings in Intestinal Neuronal Dysplasia Type B: An Advance towards Its Definition as a Disease"

_life, 2023, doi:10.3390/life13051175_

Round 1
Reviewer 1 Report
1. In the first paragraph, IND-B B should be corrected (B is described twicely).
2. The main disadavntage is the presence of a lot of limitations indicated in the discussion section. They are too serious to accept this article.
Author Response
Dear reviewer,
We would like to thank you for the relevant comments and constructive criticisms.
We have included the answers to your comments, point by point, below.
Modifications made according to the reviewers' suggestions are highlighted in blue in the revised version of the manuscript.
Thank you very much in advance for your kind attention,
Sincerely yours,
# 1) In the first paragraph, IND-B B should be corrected (B is described twicely).
We corrected this point in the first paragraph. Thank you for the note.
# 2) The main disadvantage is the presence of a lot of limitations indicated in the discussion section. They are too serious to accept this article.
We agree that our study has limitations presented and discussed in the Discussion section. However, most of these limitations are related to Intestinal Neuronal Dysplasia type B (IND-B), a rare condition with many doubts regarding its etiopathogenesis, diagnosis, and treatment. In this sense, we understand that our study contributes in an unprecedented way by demonstrating for the first-time association between the clinical signs and symptoms and the alterations found in the histopathological analysis of the rectal samples of patients with IND-B. This result significantly contributes to understanding the pathogenesis of IND-B, characterizing it as a disease, and justifying future studies to improve diagnostic and therapeutic aspects.

Reviewer 2 Report
Thank you to the authors for this study.
The study aims to correlate the symptoms and diagnosis. However, as stated by the authors, the lack of a control group is an important limitation.
The statistical tools used are quite impressive, however, they suffer from the limited number of patients. While this is mostly due to the rarity of the disease, it may cause a misinterpretation.
Author Response
Dear reviewer,
We want to thank you for the pertinent comments and constructive criticisms.
We have included the answers to your comments, point by point, below.
Modifications made according to the reviewers' suggestions are highlighted in blue in the revised version of the manuscript.
Thank you very much in advance for your kind attention,
Sincerely yours,
# 1) The study aims to correlate the symptoms and diagnosis. However, as stated by the authors, the lack of a control group is an important limitation.
We agree that the lack of a control group is a limitation of our study. This is the main limitation of the diagnostic studies regarding Intestinal Neuronal Dysplasia type B. Obtaining intestinal samples from healthy children without intestinal symptoms for use as controls has ethical limitations due to the need for invasive procedures to get tissue samples. In this sense, the statistical model used in our study mitigated this limitation, making it possible to precisely assess the presence of an association between the histopathological characteristics in the rectal samples and the symptoms presented by these same patients without making comparisons with a control group.
We have added information about these limitations in the discussion section.
# 2) The statistical tools used are quite impressive, however, they suffer from the limited number of patients. While this is mostly due to the rarity of the disease, it may cause a misinterpretation.
Intestinal Neuronal Dysplasia type B is a rare disease, with an estimated prevalence of 1:7,500 newborns. In addition, the uncertainties related to its definition and diagnostic histopathological criteria certainly influence the prevalence estimates, which may be even lower. Thus, although our sample is numerically limited, it is one of the most significant described in studies that evaluated pathophysiological aspects of IND-B.
We have added information about these limitations in the discussion section.

Reviewer 3 Report
I believe this is an excellent paper looking at a rare disease with robust sample size and a novel statistical method to me factor rotation analysis comparing clinical to histological factors
recommendations/questions after some supplemental statistical reading
Fig 2 the matrix plot seems to be the heart of the paper could you expand the explanation for clinical readers as to its meaning in the discussion, For instance are factors in the upper right quadrant more significant or more positively correlated and is distance from 0 mean a higher significance or correlation This was my impression but i think many readers will struggle to understand the graph who do not have a strong statistical background
2 Naive question was a scree plot analysis done to rank relevant factor if so I think it should be included in the paper
Author Response
Dear reviewer,
We want to thank you for the pertinent comments and constructive criticisms.
We have included the answers to your comments, point by point, below.
Modifications made according to the reviewers' suggestions are highlighted in blue in the revised version of the manuscript.
Thank you very much in advance for your kind attention,
Sincerely yours,
# 1) Fig 2 the matrix plot seems to be the heart of the paper could you expand the explanation for clinical readers as to its meaning in the discussion, For instance are factors in the upper right quadrant more significant or more positively correlated and is distance from 0 mean a higher significance or correlation This was my impression but i think many readers will struggle to understand the graph who do not have a strong statistical background
We added more detailed explanations about the graph in the results and discussion sections. We hope that these modifications make this interpretation more suitable for readers.
# 2) Naive question was a scree plot analysis done to rank relevant factor if so I think it should be included in the paper
Because in our study, the number of variables is greater than the number of observations, we opted to determine only two factors, with 28% of variation explained by Factor 1 and 17% for Factor 2, totaling 45% of variation explained by these factors.

Round 2
Reviewer 1 Report
Authors changed the part of manuscript according to my first recommendation.
However, the main limitations are still present.